# Development of Training Materials for Pathologists to Provide Machine Learning Validation Data of Tumor-Infiltrating Lymphocytes in Breast Cancer

**DOI:** 10.3390/cancers14102467

**Published:** 2022-05-17

**Authors:** Victor Garcia, Katherine Elfer, Dieter J. E. Peeters, Anna Ehinger, Bruce Werness, Amy Ly, Xiaoxian Li, Matthew G. Hanna, Kim R. M. Blenman, Roberto Salgado, Brandon D. Gallas

**Affiliations:** 1Division of Imaging, Diagnostics, and Software Reliability, Office of Science and Engineering Laboratories, Center for Devices and Radiological Health, US Food and Drug Administration, Silver Spring, MD 20993, USA; katherine.elfer@fda.hhs.gov (K.E.); brandon.gallas@fda.hhs.gov (B.D.G.); 2Cancer Prevention Fellowship Program, Division of Cancer Prevention, National Cancer Institute, National Institute of Health, Shady Grove, MD 20850, USA; 3Department of Pathology, Sint-Maarten Hospital, 2800 Mechelen, Belgium; dieter-peeters@telenet.be; 4Department of Biomedical Sciences, University of Antwerp, 2020 Antwerp, Belgium; 5Histopathology, Imaging and Quantification Unit, CellCarta NV, 2020 Antwerp, Belgium; 6Department of Genetics and Pathology, Laboratory Medicine, Region Skåne, Lund University, 22185 Lund, Sweden; anna.ehinger@med.lu.se; 7Inova Health System Department of Pathology, Falls Church, VA 22042, USA; bruce.werness@inova.org; 8Arrivebio LLC, San Francisco, CA 94111, USA; 9Department of Pathology, Massachusetts General Hospital, Boston, MA 02114, USA; aly1@partners.org; 10Department of Pathology and Laboratory Medicine, Emory University, Atlanta, GA 30322, USA; bill.li@emory.edu; 11Department of Pathology and Laboratory Medicine, Memorial Sloan Kettering Cancer Center, New York, NY 10065, USA; hannam@mskcc.org; 12Department of Internal Medicine, Section of Medical Oncology, School of Medicine and Yale Cancer Center, Yale University, New Haven, CT 06511, USA; kim.blenman@yale.edu; 13Department of Computer Science, School of Engineering and Applied Science, Yale University, New Haven, CT 06511, USA; 14Division of Research, Peter Mac Callum Cancer Centre, Melbourne, VIC 3000, Australia; roberto@salgado.be; 15Department of Pathology, GZA-ZNA Hospitals, 2020 Antwerp, Belgium

**Keywords:** tumor-infiltrating lymphocytes, pathologist training/education, expert panel, validation dataset, biomarker

## Abstract

**Simple Summary:**

The High Throughput Truthing project aims to develop a dataset of stromal tumor-infiltrating lymphocytes (sTILs) density evaluations in hematoxylin and eosin-stained invasive breast cancer specimens fit for a regulatory purpose. After completion of the pilot study, the analysis demonstrated inconsistencies and gaps in the provided training to pathologists. Select regions of interest (ROIs) were reviewed by an expert panel, who provided annotations and commentary on the challenges of the sTILs assessment. We used these annotations to develop a training document and reference standard for new training materials. These materials will train crowd-sourced pathologists to help create an algorithm validation dataset and contribute to sTILs evaluations in clinical practice.

**Abstract:**

The High Throughput Truthing project aims to develop a dataset for validating artificial intelligence and machine learning models (AI/ML) fit for regulatory purposes. The context of this AI/ML validation dataset is the reporting of stromal tumor-infiltrating lymphocytes (sTILs) density evaluations in hematoxylin and eosin-stained invasive breast cancer biopsy specimens. After completing the pilot study, we found notable variability in the sTILs estimates as well as inconsistencies and gaps in the provided training to pathologists. Using the pilot study data and an expert panel, we created custom training materials to improve pathologist annotation quality for the pivotal study. We categorized regions of interest (ROIs) based on their mean sTILs density and selected ROIs with the highest and lowest sTILs variability. In a series of eight one-hour sessions, the expert panel reviewed each ROI and provided verbal density estimates and comments on features that confounded the sTILs evaluation. We aggregated and shaped the comments to identify pitfalls and instructions to improve our training materials. From these selected ROIs, we created a training set and proficiency test set to improve pathologist training with the goal to improve data collection for the pivotal study. We are not exploring AI/ML performance in this paper. Instead, we are creating materials that will train crowd-sourced pathologists to be the reference standard in a pivotal study to create an AI/ML model validation dataset. The issues discussed here are also important for clinicians to understand about the evaluation of sTILs in clinical practice and can provide insight to developers of AI/ML models.

## 1. Introduction

Tumor-infiltrating lymphocytes (TILs) are prognostic and predictive biomarkers in triple negative breast cancer (TNBC) [1,2,3,4,5,6,7,8]. TILs densities in primary tumor specimens of patients that do or do not receive (neo)adjuvant chemotherapy demonstrate positive correlations with patient outcomes [7,8,9,10,11]. Understanding this relevance, incorporating the TILs assessment into standard clinical practice is strongly considered and actively endorsed by international clinical and pathology organizations [12,13,14]. Guidelines for standardized TILs assessment and educational materials to support researchers and pathologists to score this biomarker have been developed by the International Immuno-Oncology Biomarker Working Group (the Working Group) on Breast Cancer [15,16].

Anticipating the influx of artificial intelligence and machine learning algorithms (AI/ML) to assess TILs [17,18,19,20,21,22], we began the High Throughput Truthing (HTT) project in collaboration with an international team of pathologists, clinical scientists, and leadership from the Working Group [23]. Our goal is to create a dataset of digital slide data with pathologist annotations for the validation of computational pathology models (e.g., AI/ML) for stromal TILs (sTILs) assessment that will be fit for a regulatory purpose as a medical device development tool [24].

We focus our efforts on the stromal TILs assessment in accordance with the recommendations from the Working Group [15]. The TILs assessment requires preserved tissue, either core biopsies prior to neoadjuvant therapy or full sections, and is applicable to both primary and metastatic solid tumors [4,15]. The TILs assessment can be performed in both the stromal and intratumoral (also called intra-epithelial) tissue compartments. However, when using hematoxylin and eosin (H&E)-stained sections of invasive breast carcinoma, intratumoral TILs are more heterogenous and difficult to observe without additional staining. In addition, sTILs measurements provide the same information as those of intratumoral TILs while being a more reproducible measurement [15]. We prioritize core biopsies of the primary tumor, as metastatic disease is an area of current research [4,25,26]. Our annotations only include estimates of the density of sTILs in regions of interest (ROIs).

We recently completed a pilot study to collect sTILs from pathologists and summarized the methods and tools of the pilot study [23]. The pilot study will inform development of a pivotal study that will generate the algorithm validation dataset. The pilot study recruited board-certified pathologists and pathology residents and offered training on the sTILs interpretation: the guidelines on sTILs evaluation [15] and a video tutorial and corresponding presentation about sTILs evaluation, the project, and using the platforms [27]. We have since updated the training to include a video produced by the Working Group [28,29,30].

Analyzing the pilot study data, we observed notable pathologist variability in sTILs estimates. To understand and address this variability, we established an expert panel to review a subset of ROIs. We aggregated, consolidated, and utilized their comments and annotations to create additional training materials for the pivotal study.

In this manuscript, we describe the expert panel sessions, the annotations collected by our experts in comparison to those from the pilot study, and the sTILs assessment pitfalls encountered in these ROIs. We are not exploring AI/ML performance in this work. Instead, we are creating materials that will train crowd-sourced pathologists to be the reference standard in our pivotal study to create an algorithm validation dataset. The issues discussed here about the evaluation of sTILs in clinical practice are also important for clinicians to understand and can provide insight to developers of AI/ML models.

## 2. Materials and Methods

### 2.1. Pilot Study

Our pilot study pathologist annotation data are publicly available from our GitHub repository [31]. We recruited twenty-nine pathologists through conferences and pathology communities. These are the “crowd-sourced” pathologists. Interested pathologists were directed to the project hub [32] that detailed instructions for registration, training, and participation. The registration collected board-certification information and experience; these data can also be found in our GitHub repository [31]. The range of self-reported experience started with residents and maxed out with board-certified pathologists with 40 years of clinical practice. Some pathologists did not report their experience. The training was not monitored but indicated that participants were required to watch a video webinar on the sTILs assessment [33] and read the guidelines from the Working Group [15].

The pilot study produced a total of 7373 sTILs density estimates for 640 unique ROIs. Pathologists could use optical or digital modalities: a light microscope system (eeDAP [34,35]) and two digital whole slide image viewing and annotation platforms (caMicroscope [36] and Path Presenter [37]). From 64 H&E-stained slides of breast cancer biopsies, a collaborating pathologist selected ten unique ROIs of varying morphology from each slide according to the protocol described previously [23]. Slides were scanned on a Hamamatsu Nanozoomer 2.0-RS C10730 series at 40x equivalent magnification (0.23 µm/pixel). The analysis in this manuscript is limited to data collected on the caMicroscope digital platform from February 2020 to May 2021, because most of the data were collected with this modality. The code to generate this analysis can be found in the https://github.com/DIDSR/HTT (accessed on 6 May 2022). To improve and finalize our data collection methods, we describe and assess the technical workflows and explore the PathPresenter data in a separate paper.

### 2.2. Collected Annotations

We captured three data elements for each ROI: ROI label, percent tumor-associated stroma, and sTILs density. The ROI label is a qualitative variable that describes the tissue within the ROI as either “Intra-Tumoral Stroma”, “Invasive Margin”, “Tumor with No Intervening Stroma”, or “Other Regions.” The “Intra-Tumoral Stroma” and “Invasive Margin” tissues are regions where tumor-associated stroma and sTILs can be found; however, not all tumor-associated stroma contain sTILs. “Tumor with No Intervening Stroma” and “Other Regions” are regions where there is no tumor-associated stroma, and, by definition, there can be no sTILs. Given these associations, the ROI label offers an additional opportunity to evaluate whether an algorithm is estimating the sTILs density in the proper regions. The labels specifying that an ROI is evaluable for sTILs include “Intra-Tumoral Stroma” and “Invasive Margin”, while the labels that indicate an ROI is not evaluable for sTILs are “Tumor with No Intervening Stroma” and “Other Regions”.

The variable percent tumor-associated stroma is the percentage of tumor-associated stroma present within the ROI and is calculated as:(1)$$\mathrm{Percent}\;\mathrm{Tumor}-\mathrm{Associated}\;\mathrm{Stroma}=\left(\frac{\mathrm{Area}\;\mathrm{of}\;\mathrm{Tumor}-\mathrm{Associated}\;\mathrm{Stroma}}{\mathrm{Area}\;\mathrm{of}\;\mathrm{Entire}\;\mathrm{ROI}}\right)\times 100\%.$$

This variable represents the visually estimated percent of the entire ROI (including empty space) covered by tumor-associated stroma; the compartment in which the sTILs density is evaluated. The percent of tumor-associated stroma is not expected to be reported clinically. However, segmenting the stroma is an important step in estimating the sTILs density. As such, we ask for the percent of tumor-associated stroma to remind the pathologist about the segmentation step. The data can also be used to assess an AI/ML model’s ability to identify tumor-associated stroma, a component of the sTILs density.

The sTILs density is the percentage of the TILs area within tumor-associated stroma and is calculated as
(2)$$\mathrm{sTILs}\;\mathrm{Density}=\left(\frac{\mathrm{Area}\;\mathrm{of}\;\mathrm{Tumor}-\mathrm{Infiltrating}\;\mathrm{Lymphocytes}}{\mathrm{Area}\;\mathrm{of}\;\mathrm{Tumor}-\mathrm{Associated}\;\mathrm{Stroma}}\right)\times 100\%$$

Both the sTILs density and percent tumor-associated stroma assessments will be recorded as a continuous variable ranging from 0 to 100%. TILs are limited to lymphocytes and plasma cells. Granulocytes, dendritic cells, and macrophages are not considered in the quantitative assessment [15].

### 2.3. Selecting Regions of Interest for the Expert Panel Sessions

We selected ROIs from the pilot study based on their mean sTILs score, sTILs variance, and ROI label entropy. The sTILs means and variances are averages over readers for each ROI; each ROI must have at least two pathologist scores for a variance to be calculated. For a given ROI, we also calculated the entropy of the pathologist labels:(3)$$\mathrm{Label}\;\mathrm{Entropy}=-\sum_{i=1}^{4}p_{i}\log\left(p_{i}\right),\;$$
where i indexes the label, and $p_{i}$ is the fraction of readers that labeled the case with label i. Entropy is a measure of variance for categorical data [38,39]; it captures both the number of different labels given to an ROI and the frequency of the labels. The entropy for an ROI for which all readers give the same label will be zero. The entropy then increases as the distribution of labels is more evenly spread among all the labels.

When selecting the ROIs for the expert session, only those ROIs with a calculated variance were included, which reduced the number of available ROIs from 640 to 570. Figure 1 shows a plot of the sTILs density mean and variance for each ROI. We stratified our sampling into three sTILs density bins: low infiltration = “10% or less”, moderate infiltration = “greater than 10% to 40%”, and high infiltration = “greater than 40%”. The thresholds for these bins appear as dashed vertical lines in Figure 1. These thresholds were recommended by our clinical experts to split the range into possible patient management bins [40,41]. We then selected cases with the highest variance and entropy and lowest variance and entropy using a 2:1 high–low ratio for a total of 72 ROIs.

Examples of selected ROIs are found in Figure 2 and Figure 3. For contrast, the cases with the highest and lowest variance and entropy within the “low infiltration” (less than or equal to 10%) and “high infiltration” (greater than 40%) sTILs density bins are shown. For these example ROIs, Table 1 lists the summarized annotations, and Table 2 contains a breakdown of the frequency of ROI labels. These tables also show the corresponding label entropy for each ROI. Comparing Figure 3A and Figure 3B, the entropy decreases from 1.1 to 0 (Table 1) as fewer different ROI labels are chosen (Table 2).

We split the 72 ROIs into two batches: Training Batch I and Training Batch II. Training Batch I is intended to be a training set for a test with feedback, and Training Batch II is to be used for a proficiency test.

### 2.4. Expert Panel Sessions

The expert panel consisted of seven board-certified pathologists and one translational scientist; all are project collaborators and trained in sTILs assessment. The board-certified pathologists have 3–33 years of clinical experience, and the translational scientist is an immunologist and clinical chemist working on breast pathology, immunology, and drug development for over 10 years. We held eight recorded, one-hour virtual sessions for the expert panel members to discuss each selected ROI regarding their sTILs assessment. At least three expert panel members participated in each session. After the discussions, the experts revisited the ROIs and recorded their annotations using the digital platform caMicroscope [36].

The semi-structured expert panel sessions encouraged discussion of diverse viewpoints on their approach to sTILs assessment. The sessions were conducted via Zoom with a facilitator sharing their screen showing ROIs with the caMicroscope interface. The experts silently considered each ROI while deciding on the ROI label, the percent tumor-associated stroma, and the sTILs density. In some cases, an expert asked the facilitator to pan and zoom to other areas of the image to better understand the context of the ROI. Each pathologist then revealed their annotations. They also commented on the ROI features influencing their assessment and how they arrived at their annotations. The majority of Training I ROIs were scored by the experts after completing the group review. All the Training II ROIs were scored before the group review.

Following the sessions, we compiled, analyzed, and consolidated the experts’ scores, comments, and pitfalls. One expert pathologist did not complete annotations on all the selected ROIs; their annotations were not included in the analysis. We limited the analysis to include the six experts who completed annotations on all selected ROIs.

## 3. Results

Figure 4 shows a graphical comparison of the change in sTILs density variance between the crowd pathologists and the experts plotted using an ROI’s mean sTILs density as determined by the crowd. The majority of sTILs density variances from the expert panel were smaller than the variances from the crowd-sourced annotations. There is one outlier ROI from the experts that has a variance of 2700. For this ROI, three experts assessed the ROI label as “Intra-Tumoral Stroma” with sTILs densities of 0, 90, and 90. These experts all believed there was high percent tumor-associated stroma (90, 90, 99). The other three experts assessed the ROI as “Other Regions” for which the sTILs densities are not defined and are not collected. Summary statistics of the sTILs density variances are described in Table 3. The select ROIs (Crowd–Select) have higher variability compared to the full dataset (Crowd–All), which is expected; the experts have less variability than the crowd.

Figure 5 shows a graphical comparison of the change in entropy between the crowd pathologists and the expert panel with ROIs matched on the mean sTILs density as determined by the crowd pathologists. The majority of sTILs density entropies from our experts decreased in comparison to the crowd-sourced annotations. The summary statistics of the ROI label entropies are described in Table 4. The labels from crowd on the selected ROIs (Crowd–Select) have higher entropy compared to the full dataset (Crowd–All). Additionally, the labels from the experts have less entropy than the labels from the crowd on the selected ROIs (Crowd–Select). For the expert panel, the median entropy is 0.00 for all bins, which means that the majority of entropy values are zero; the experts were largely in agreement in determining the ROI label. The lower median entropies in the expert panel (Table 4) reflect the decreased frequency of the multiple ROI labels, as summarized in Table 5.

From the expert panel commentary, we identified recurring attributes that complicated the sTILs assessment and refer to them as pitfalls. We have compiled these pitfalls into a reference document to add to the training materials. Generally, the instructions to pathologists are to know about these pitfalls and consider them when performing their sTILs assessment. These pitfalls can be grouped into two main categories: pitfalls related to estimating the percent of tumor-associated stroma and pitfalls related to estimating sTILs density. The etiologies of these categories include the slide preparation process, limitations of H&E staining, and the pathologists’ assessment. The pitfalls are listed below and summarized in Table 6.

The percent of tumor-associated stroma assessment had four identified pitfalls:Not all mesenchymal tissue should be considered tumor-associated stroma. For the purposes of sTILs assessments, tumor-associated stroma is defined as the reactive stroma composed of fibroblasts, newly formed vessels, collagen fibers, and extracellular matrix surrounding invasive carcinoma cells and cell nests. Pre-existing normal structures, such as adipose tissue, blood vessels, or nerves, are excluded from the area segmented as tumor-associated stroma. Areas of necrosis and fibrin are also excluded.The percent of tumor-associated stroma is calculated with respect to the area of the entire ROI, as previously described. Vessel lumens, adipose tissue, and negative (empty) space should be included in the total ROI area, the denominator of the percent tumor-associated stroma equation. The numerator is only tumor-associated stroma.Variations in tumor cell morphology can make it difficult to distinguish stroma from tumor. Tumor cell cytoplasmic eosinophilia can be similar to that of adjacent stroma and cause difficulty in distinguishing these two tissue types. Additional stains may be useful in these scenarios.Carcinoma in situ and benign glandular elements entrapped within the tumor area, including intact terminal duct lobular units, should be excluded from the numerator when calculating the percent of tumor-associated stroma.The sTILs density score assessment had four identified pitfalls:Cells with small/pyknotic nuclei and/or perinuclear clearing can be difficult to categorize as macrophages, tumor cells, plasma cells, or lymphocytes. This may occur with invasive lobular carcinoma or in cases of suboptimal tissue fixation. Additional stains may be helpful.Non-lymphoid cells that may be confused for lymphocytes include cross-sectionally cut fibroblasts and tumor cells, particularly if low grade and/or degenerated. Sometimes, cancer cell nuclei are hyperchromatic, due to crush artifacts, overstaining, and/or poor fixation, and can be confused for lymphocytes.Error in the percent tumor-associated stroma can lead to inflated or deflated sTILs scores. Stroma may be obscured by dense populations of cells and may be incorrectly excluded from the sTILs evaluation. A lower estimated percent tumor-associated stroma could substantively affect the sTILs score.When tumor cells are sparsely distributed throughout the ROI, it may be more challenging to accurately quantitate the sTILs density and percent tumor-associated stroma.

The expert sessions also revealed glimpses into the cognitive processes used by pathologists to complete their assessment. Some would mentally relocate tissue to a portion of the ROI, while others mentally overlaid geometric shapes to estimate areas. One pathologist used a “forced binary choice” approach to narrow down area estimates, e.g., <50% vs. ≥50%, before concluding their estimates.

## 4. Discussion

In this work, we have described our efforts to create additional training materials to improve the quality of the upcoming pivotal study. Through an expert panel, we generated reference annotations to educate professionals on the sTILs assessment. We selected ROIs such that the variability of the crowd pathologists was larger for the selected ROIs compared to all the ROIs in the pilot study. The variability from the expert panel was lower than the variability from the crowd thanks to their dedication to the task, the discussions with their peers, and their use of freely available training schemes on the website of the Working Group [16].

The selected ROIs and expert annotations created in this work will be used to create two sets of data: a training set for a test with feedback and a second set that will be used in a proficiency test. The test with feedback is a new workflow created on the same data-collection platforms as the pilot study. When a pathologist enters their sTILs assessment and clicks “Save”, they will be shown the expert panel’s sTILs assessments, comments, and pitfalls for that ROI. The feedback is presented while the ROI is still visible, allowing the pathologist to study and reflect on the image, their initial assessment, and information from the experts. The immediacy of this feedback will facilitate participants’ performance improvement, as demonstrated in the educational literature [42].

The proficiency test will require future study participants to demonstrate their ability in the sTILs assessment and perform above a specified metric, which will be determined from the experts’ annotations. In doing so, pivotal study participants will demonstrate that they can perform the sTILs assessment with a similar degree of proficiency as the experts. With this addition, we anticipate less variability of sTILs density estimates in the pivotal study, just as the experts had less variability than the crowd pathologists, and a higher quality validation dataset.

The test with feedback and the proficiency test will be mandatory training materials combined with the original training materials [15,27,28,29,30,33]. We did not monitor the training for the pilot study, and it is likely that some study participants did not do the training. For the pivotal study, the pathologists will have to achieve a level of proficiency with their sTILs assessments to participate.

As a result of this work, we are also changing the ROI label data element. During algorithm validation, the ROI label’s intention is to assess whether or not an algorithm correctly determines whether an ROI should be considered for sTILs assessment. As seen in Table 5, there were two ROI labels used most frequently: “Intra-Tumoral Stroma” and “Other Regions”. Considering the intention of the ROI label, the observed frequencies, and the feedback of our experts, instead of characterizing among four types of tissues, the new ROI label data elements will reflect whether the tissue within the ROI should be considered for the sTILs assessment. The new ROI label options will be “Evaluable for sTILs” and “Not evaluable for sTILs”. This change will decrease ambiguity of the data element and facilitate binary analysis methods after completion of the pivotal study.

Our work describes various difficulties that participants may encounter during their assessment of sTILs, as summarized in Table 6, which are similar to pitfalls described by Kos et al. [43,44]. In their work, the authors discuss various pitfalls related to technical factors, such as out-of-focus scanned WSIs; scoring the wrong area of cell type; when there is low amount of stroma; and how to approach a heterogeneity of sTILs densities within the tumor. Examples of similar pitfalls are that only lymphocytes and plasma cells are included in the sTILs evaluation, a crush artifact can obfuscate the sTILs assessment, and lymphocytes associated with benign glandular tissue are excluded from the sTILs assessment. Our document highlights specific pitfalls encountered in our dataset, while the Kos et al. work includes factors beyond our dataset with clinical recommendations. For additional information on the sTILs assessment, the Working Group has more training materials and a freely available training tool for the community on their website [16].

Our work not only offers opportunities to improve the education of study participants but offers insight for algorithm developers. As AI/ML pitfalls in the sTILs assessment become better understood, the pathologist commentary on pitfalls related to the sTILs assessment can inform challenges in validating an AI/ML algorithm. For example, an AI/ML tool validated using the classical ductal phenotypes may find it difficult to identify the cancer cells in an ROI with tumor showing apocrine features because the tumor cells are as eosinophilic as the stroma. Similarly, if AI/ML models are not trained with proper ground truth, they may confuse lymphoid aggregates, such as tertiary lymphoid structures, for TILs.

Limitations of our work include the ROI selection criteria as well as the semi-structured discussion-based review and data-collection process by the expert panel. We selected ROIs from among cases that had a calculated variance. Cases with ROI labels of only “Other Region” and “Tumor with No Intervening Stroma” would not have sTILs density scores or variances and were excluded from the selected ROIs. This affected the selection of the high and low entropy cases and may have affected the pitfalls we identified. Regarding the expert panel review and data-collection process, we did not follow a strict method [45]. Therefore, the conclusions drawn from the analysis of improvements in the sTILs density variances are observational. Our work was not intended to yield unbiased consensus data or study the impact of expert training sessions on sTILs density variances. Our goal was to understand pathologist variability and improve training materials.

## 5. Conclusions

In summary, through an expert panel, we created additional training materials for study participants. A training set and proficiency test have been added as mandatory components to the training protocol. We also created a reference document from pitfalls encountered during the sTILs assessment that will be used as part of the feedback in the training set. Using these improved training metrics, we set higher standards in the proficiency required for participation in our pivotal study. This will lead to decreased variability in pivotal study annotations, which in turn, translates to a better machine learning validation dataset.

## Figures and Tables

**Figure 1 cancers-14-02467-f001:**
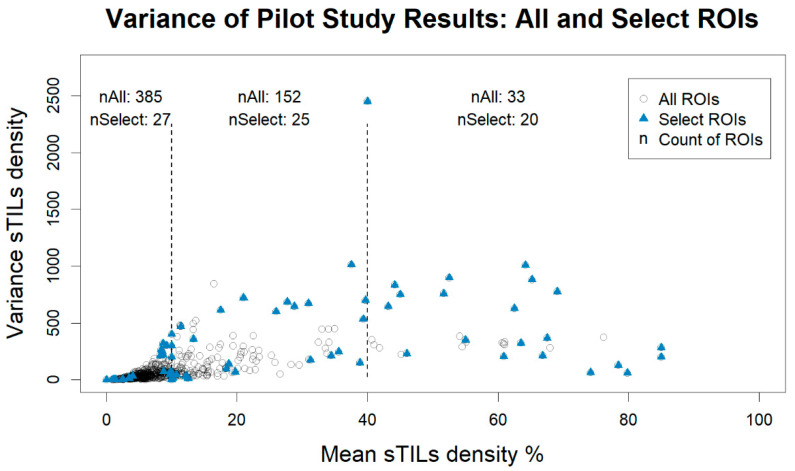
Plot of the stromal tumor-infiltrating lymphocytes (sTILs) variance vs. mean density for the pilot study. Plotted data include all data collected on the digital platform caMicroscope (All ROIs, circles) and the same data restricted to regions of interest (ROIs) selected for further evaluation by the expert panel (Select ROIs, solid triangles). ROIs were selected to be distributed equally across low, moderate, and high infiltration levels based on the clinically recommended thresholds of 10% and 40%, represented by the vertical dashed lines. Within each density bin, ROIs with the highest and lowest variance and entropy were selected for expert panel review. nAll and nSelect are the respective counts of All ROIs and Select ROIs within each bin: “≤ 10%” nAll = 385, nSelect = 27; “10% < % ≤ 40%” nAll = 152, nSelect = 25; “> 40%” nAll = 33, nSelect = 20.

**Figure 2 cancers-14-02467-f002:**

Example regions of interest (ROIs) selected by pathologist variance of stromal tumor-infiltrating lymphocytes (sTILs). Each ROI is 500 µm × 500 µm and has a 100 µm bar for scale. The collected annotations for these ROIs are listed in Table 1 and Table 2. (**A**) High sTILs variance, mean sTILs density ≤ 10% (LE10). (**B**) Low sTILs variance, mean sTILs density ≤ 10% (LE10). (**C**) High sTILs variance, mean sTILs density > 40% (GT40). (**D**) Low sTILs variance, mean sTILs density > 40% (GT40).

**Figure 3 cancers-14-02467-f003:**
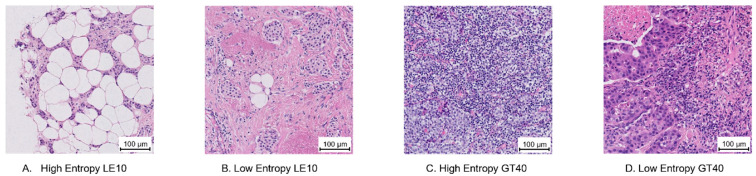
Example regions of interest (ROIs) selected by pathologist entropy of the ROI label. Each ROI is 500 µm × 500 µm and has a 100 µm bar for scale. The collected annotations for these ROIs are listed in Table 1 and Table 2. (**A**) High ROI label entropy, mean sTILs density ≤ 10% (LE10). (**B**) Low ROI label entropy, mean sTILs density ≤ 10% (LE10). (**C**) High ROI label entropy, mean sTILs density > 40% (GT40). (**D**) Low ROI label entropy, mean sTILs density > 40% (GT40).

**Figure 4 cancers-14-02467-f004:**
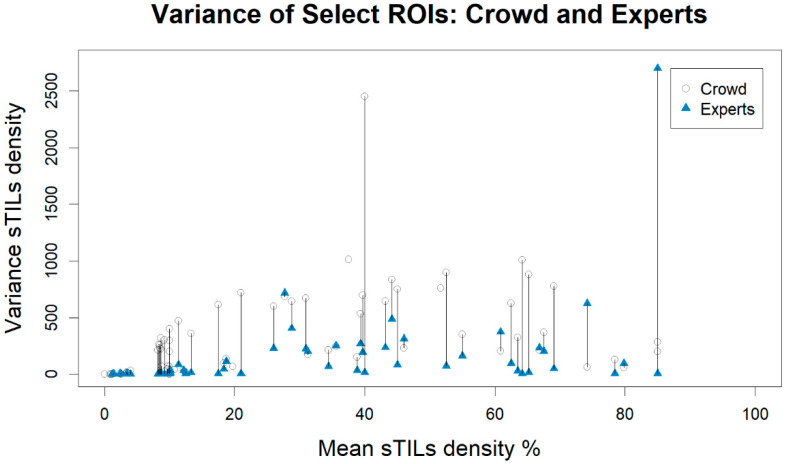
Plot of the variance vs. mean stromal tumor-infiltrating lymphocytes (sTILs) density for the regions of interest (ROIs) selected for the expert panel. ROIs are matched on their unique case identifiers and plotted with the mean sTILs density as determined by the crowd pathologists. Variances belonging to the same ROI are connected by straight lines. Cases in which there was no calculable variance from the experts’ assessment are represented by an open circle (Crowd) without a connected blue triangle (Experts).

**Figure 5 cancers-14-02467-f005:**
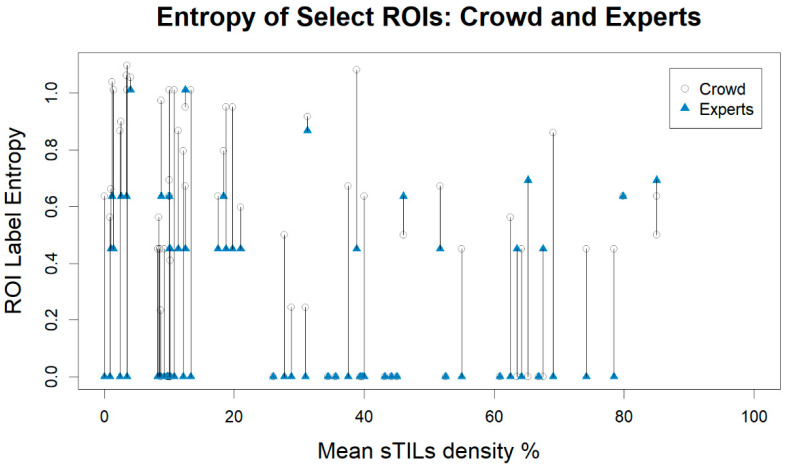
Plot of the region of interest (ROI) label entropy vs. mean stromal tumor-infiltrating lymphocytes (sTILs) density for the ROIs selected for the expert panel. ROIs are matched on their case identifiers and plotted with their mean sTILs score density as determined by the crowd pathologists. ROI label entropies belonging to the same ROI are connected by straight lines.

**Table 1 cancers-14-02467-t001:** Summary statistics of collected annotations from crowd pathologists for the example regions of interest (ROIs) in Figure 2 and Figure 3 within the stromal tumor-infiltrating lymphocytes (sTILs) density bins of “less than or equal to 10%” (LE10) and “greater than 40%” (GT40). For the High Entropy LE10 (Figure 3A) case, there is a tie for the Majority Label; the multiple labels are separated by “*AND*”.

Figure	Figure Description	Mean sTILs Density	Variance	Majority Label	Entropy
2A	High Variance LE10	10	400	Intra-Tumoral Stroma	1.01
2B	Low Variance LE10	0	0	Other Regions	0.64
2C	High Variance GT40	64.2	1008.2	Intra-Tumoral Stroma	0.45
2D	Low Variance GT40	79.83	58.97	Intra-Tumoral Stroma	0.64
3A	High Entropy LE10	3.5	9.67	Intra-Tumoral Stroma *AND* Invasive Margin *AND* Tumor with No Intervening Stroma	1.1
3B	Low Entropy LE10	9.75	70.79	Intra-Tumoral Stroma	0
3C	High Entropy GT40	69.08	775.9	Intra-Tumoral Stroma	0.86
3D	Low Entropy GT40	66.83	212.17	Intra-Tumoral Stroma	0

**Table 2 cancers-14-02467-t002:** Frequency of region of interest (ROI) labels of collected annotations from crowd pathologists for the example ROIs in Figure 2 and Figure 3 within the stromal tumor-infiltrating lymphocytes (sTILs) density bins of “less than or equal to 10%” (LE10) and “greater than 40%” (GT40).

Figure	Figure Description	Invasive Margin	Intra-Tumoral Stroma	Tumor with No Intervening Stroma	Other Regions
2A	High Variance LE10	1	3	2	0
2B	Low Variance LE10	0	2	0	4
2C	High Variance GT40	0	5	1	0
2D	Low Variance GT40	2	4	0	0
3A	High Entropy LE10	2	2	2	0
3B	Low Entropy LE10	0	8	0	0
3C	High Entropy GT40	2	10	3	0
3D	Low Entropy GT40	0	6	0	0

**Table 3 cancers-14-02467-t003:** Variance summary statistics of the crowd annotations from all regions of interest (ROIs) scored using the caMicroscope modality (Crowd-All), the crowd annotations of the selected ROIs (Crowd-Select), and the expert panel’s annotations of the selected ROIs. Data are grouped by the mean stromal tumor-infiltrating lymphocytes (sTILs) density bin and reported as the median variance and (interquartile range).

	All Densities	≤10%	10% < % ≤ 40%	>40%
Crowd-All	48.10 (20.58–110.31)	30.70 (15.07–59.50)	111.50 (56.30–245.13)	324.55 (278.17–627.50)
Crowd-Select	212.24 (39.33–549.50)	44.67 (4.05–225.28)	246.80 (67.58–646.18)	358.75 (210.17–762.73)
Experts	14.17 (4.23–178.67)	3.07 (0.98–4.32)	70.00 (14.17–224.17)	96.67 (39.42–275.03)

**Table 4 cancers-14-02467-t004:** Entropy summary statistics of the labels from all regions of interest (ROIs) scored using the caMicroscope modality (Crowd-All), the crowd labels of the selected ROIs (Crowd–Select), and the expert panel’s labels of the selected ROIs. Data are grouped by the mean stromal tumor-infiltrating lymphocytes (sTILs) density bin and reported as the median entropy and (interquartile range).

	All Densities	≤10%	10% < % ≤ 40%	>40%
Crowd-All	0.23 (0.00–0.45)	0.23 (0.00–0.41)	0.24 (0.00- 0.50)	0.00 (0.00–0.45)
Crowd-Select	0.56 (0.00–0.86)	0.64 (0.45–0.99)	0.64 (0.24–0.92)	0.45 (0.00–0.52)
Experts	0.00 (0.00–0.45)	0.00 (0.00–0.45)	0.00 (0.00–0.45)	0.00 (0.00–0.50)

**Table 5 cancers-14-02467-t005:** Frequency of the calculated majority region of interest (ROI) labels. Counts are grouped as all ROIs scored using caMicroscope (Crowd–All), selected ROIs annotated by the crowd pathologists (Crowd–Select), and the selected ROIs annotated by the experts (Experts).

Majority Label	Crowd-All	Crowd-Select	Experts
Intra-Tumoral Stroma	525 (82.03%)	54 (75%)	56 (77.78%)
Intra-Tumoral Stroma *AND* Invasive Margin	10 (1.56%)	1 (1.39%)	1 (1.39%)
Intra-Tumoral Stroma *AND* Invasive Margin *AND* Tumor with No Intervening Stroma	1 (0.16%)	1 (1.39%)	0 (0%)
Intra-Tumoral Stroma *AND* Other Regions	2 (0.31%)	0 (0%)	2 (2.78%)
Intra-Tumoral Stroma *AND* Tumor with No Intervening Stroma	4 (0.62%)	1 (1.39%)	0 (0%)
Invasive Margin	8 (1.25%)	2 (2.78%)	1 (1.39%)
Invasive Margin *AND* Other Regions	1 (0.16%)	1 (1.39%)	0 (0%)
Other Regions	80 (12.5%)	7 (9.72%)	12 (16.67%)
Tumor with No Intervening Stroma	9 (1.41%)	5 (6.94%)	0 (0%)

**Table 6 cancers-14-02467-t006:** Summary of pitfalls encountered during the stromal tumor-infiltrating lymphocytes (sTILs) assessment grouped by pitfall type. Region of interest is abbreviated as “ROI”.

Pitfall Type	Pitfall Summary
Percent of Tumor-Associated Stroma	Exclude thick-walled vessels, benign glandular elements, adipocytes, carcinoma in situ, and necrosis from the area of tumor-associated stroma
	Calculate with respect to the entire ROI area
	Variations in tumor cell morphology can make it difficult to distinguish stroma from tumor
sTILs Density Score	Cells with small/pyknotic nuclei and/or perinuclear clearing can be difficult to categorize
	Non-lymphoid cells may be confused for lymphocytes
	Error in the percent tumor-associated stroma can affect the sTILs density
	Sparsely distributed tumor cells may be more challenging to quantitate

## Data Availability

The expert panel and pilot study annotations are immediately available on this public repository: https://github.com/DIDSR/HTT (accessed on 6 May 2022). Regions of interest are available via an API demonstrated in the *getROI* function in the HTT repository. Whole slide images are available for download through caMicroscope: http://htt.camicroscope.org/(accessed on 6 May 2022).

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
