# Peer review of "Development of Training Materials for Pathologists to Provide Machine Learning Validation Data of Tumor-Infiltrating Lymphocytes in Breast Cancer"

_cancers, 2022, doi:10.3390/cancers14102467_

Round 1

Reviewer 1 Report

The authors present an interesting and very well written report addressing the somewhat confusing and unresolved machine learning approaches for tumor infiltrating lymphocytes with breast cancers. The research utilizes distinct pragmatic and standard pathologic approaches at scoring and evaluation of the tumor stomas populations in breast cancer progression. The authors go on to show how the distinct technical and conceptual issues are still required in order to implement consistent and valuable scoring of TILs in breast cancer. The authors go on to show that Machine Learning tools can assist and enhance the pathologist diagnosis of unique inflammatory breast cancers. Prior to publication I would suggest some considerations for the authors to enhance the information conveyed to their targeted audience. The greatest concern I have is cell versus tissue levels of image segmentation for classification and analysis.

Specific concerns:

  • The authors should consider discussing how different samples from the grossing room affect the reliability of TIL scoring. Distinct challenges with FNA’s vs surg path tissues should be considered.
  • Are the tools the authors consider only appropriate in primary breast tissue and not metastatic tissues? Would a breast cancer in bone have ML assisted scoring similar to that of a liver biopsy?
  • Can ML tools assist a distinct margin geometrically or can a score be developed where some tumors have clear tumor stroma boundary and other single cells invasive with high dispersal be unique for TIL classifications?
  • Does the cellularity or ECM/collagen content affect/restrict or correlate TIL scoring?

Reviewer 2 Report

Great effort has been made in this paper,  well presented,  well written 

However,  I have one issue which I believe it is important to address 

The title of the paper has mentioned Machine learning 

It would be very nice if you show how helpful the proposed protocol with ML/DL

Therefore I would expect to see any ML classification task model and how proposed protocol makes difference. Comparison with old methods.

Without doing the paper seems missing the core part.

Reviewer 3 Report

The manuscript titled “Protocol Development for Pathologists to Provide Machine Learning Validation Data of TILs in Breast Cancer” describes the authors compared the crowd pathologists and expert panel pathologists on tumor-infiltrating lymphocyte density evaluation from H&E histology data of breast cancer patients. And further, the authors made the training materials and training protocol for the study participants to decrease the variability which might be used for artificial intelligence in the future. It seems the selected crowd is at least comparable with the experts to some degree (Table 5). The followings are some concerns and comments have been pointed out that the authors may want to consider.

Concerns and Comments:

  1. Line 3 the Title: I’d suggest the authors use the full name in the title instead of TILs.
  2. Lines 6-28: Country information is missing. Even we know FDA, MA, GA, etc are in the USA. Please complete the information.
  3. Please provide the details of important information: What’s the meaning of “crowd-sourced pathologists”? Are they just very normal people? Or what’s their background? Are there any special requirements? Are there any experience requirements for expert pathologists? For example, how many years of experience? These might have a high impact on the variance and entropy distribution.
  4. Lines 53-54 Keywords: “reference standard” is not suitable for a keyword, it only appears one time in the main context line 96. “predictive maker” and “prognostic marker” are not suitable for the keywords as well. I’d suggest the authors use “biomarker”.
  5. Lines 100-102: The aim of this manuscript shown here is not well-matched with the title and the conclusions. Please modify.
  6. Line 175 Figure 1: Please list the number “n” in the figure legend as well instead of only in the image.
  7. Line 194, Table 1, and Table 2: Please add a column to the most left of each table and add the serial number to each row for easier tracking.
  8. Figure 2 and Figure 3: Please add a scale bar in each of the images.
  9. Table 3 and Table 4: the last column should be “>40%”. Please double-check.
  10. The training protocol and training materials (references 16, 18, 19, etc) are the important part of the manuscript. Please emphasize what are the improvements in the training materials and training protocol for the study participants. It might be a good side-by-side comparison to make it clearer.
  11. Lines 513-515 reference 32: Dear authors, do you think it’s good to cite a manuscript that is still ongoing preparation? And what do you think 11 references out of all 33 are including the authors in this manuscript?

Round 2

Reviewer 2 Report

comments were addressed, the dataset should be public 

Reviewer 3 Report

The following comments that the authors should consider and please seriously revise the manuscript again before publication. 

  1. Line 240: The “*AND*” here is italic, it might not have to.
  2. Line 605 reference 42: I can’t find any useful information from here. Please double-check.
